# Evidence of ubiquitous Alfvén pulses transporting energy from the photosphere to the upper chromosphere

Jiajia Liu [1], Chris J. Nelson [1,2], Ben Snow [1], Yuming Wang [3,4] & Robert Erdélyi [1,5]

The multi-million degree temperature increase from the middle to the upper solar atmosphere is one of the most fascinating puzzles in plasma-astrophysics. Although magnetic waves might transport enough energy from the photosphere to heat up the local chromosphere and corona, observationally validating their ubiquity has proved challenging. Here, we show observational evidence that ubiquitous Alfvén pulses are excited by prevalent intensity swirls in the solar photosphere. Correlation analysis between swirls detected at different heights in the solar atmosphere, together with realistic numerical simulations, show that these Alfvén pulses propagate upwards and reach chromospheric layers. We found that Alfvén pulses carry sufficient energy flux (1.9 to 7.7 kW m$^{-2}$) to balance the local upper chromospheric energy losses (~0.1 kW m$^{-2}$) in quiet regions. Whether this wave energy flux is actually dissipated in the chromosphere and can lead to heating that balances the losses is still an open question.

[1] Solar Physics and Space Plasma Research Centre (SP2RC), School of Mathematics and Statistics, The University of Sheffield, Sheffield S3 7RH, UK. [2] Astrophysics Research Centre (ARC), School of Mathematics and Physics, Queen's University, Belfast, Northern Ireland BT7 1NN, UK. [3] CAS Key Laboratory of Geospace Environment, Department of Geophysics and Planetary Sciences, University of Science and Technology of China, 230026 Hefei, Anhui, China. [4] CAS Center for Excellence in Comparative Planetology, 230026 Hefei, Anhui, China. [5] Department of Astronomy, Eötvös Loránd University, Budapest Pázmány P. sétány 1/A, H-1117, Hungary. Correspondence and requests for materials should be addressed to J.L. (email: jj.liu@sheffield.ac.uk)

Vortices are ubiquitous in the universe on a huge variety of scales, from sinking water in domestic taps to spiral galaxies[1]. The first evidence of localised swirls in the solar photosphere, which had been widely hypothesised to form as a consequence of intergranular flows[2], was obtained two decades ago[3]. Considerable efforts have since been made to find their counterparts in the higher solar atmosphere, from the perspectives of analytical theory[4], magnetohydrodynamic (MHD) simulations[5–7] and high-resolution observations[8,9]. Numerical simulations have suggested that upwardly propagating MHD waves (especially Alfvén waves) can be excited by photospheric swirls and can, in certain circumstances, carry significant amounts of energy into the middle and upper layers of the solar atmosphere[7,10,11]. However, the above scenario has yet to be confirmed by observations. Even if the propagation of these swirls from the photosphere to the chromosphere were to be confirmed, their ubiquity needs to be successfully assessed in order for them to remain a viable candidate to carry sufficient energy throughout the solar atmosphere[12].

Signatures of 14 long-lived (~10 min) solar tornadoes, which could be related to large-scale swirls (with average diameter ~1.5 Mm), have been manually identified using multi-wavelength images[13]. It was suggested that, Alfvén waves co-existing with these solar tornadoes could play an import role in local energy channelling into the solar corona. However, two crucial questions remain: firstly, given that only a relatively small number of large, long-lived swirl-related tornadoes were observed in a small field of view (FOV), are such events ubiquitous in the entire solar atmosphere? And secondly, could all swirls live long enough to actually form tornadoes? In other words, are waves or pulses more common in transporting energy in the solar atmosphere? Because we do not yet know whether pulses dissipate in a different way than waves, this latter question is fundamentally important when assessing the affinity to dissipate these perturbations.

We applied the automated swirl detection algorithm (ASDA)[14] to photospheric observations with a pixel size of ~39.2 km sampled by the Solar Optical Telescope (SOT) onboard the Hinode satellite[15], and found on average 21.5 photospheric swirls in each frame with a FOV of ~800 Mm$^2$ (suggesting a swirl population of ~1.6 × 10$^5$ in the photosphere). These swirls had an average radius and rotating speed of ~290 and ~0.9 km s$^{-1}$, respectively. Applying ASDA to photospheric observations (Fe I 6302.5 Å wideband) by the CRisp Imaging SpectroPolarimeter (CRISP) mounted on the Swedish 1-meter Solar Telescope (SST)[16] revealed similar results. The details of swirl detection using ASDA is in the methods section (Eq. 1).

Here, we report a step-forward by analysing the co-spatial and co-temporal relationship between photospheric and chromospheric swirls automatically detected by ASDA from both SOT (see the Methods section and Fig. 1) and SST (see Supplementary Note 1 and Supplementary Fig. 1) observations. Correlation analysis between swirls detected at different heights from observations of both instrument, together with realistic numerical simulation, show that ubiquitous Alfvén pulses are excited by prevalent intensity swirls in the solar photosphere. These Alfvén pulses could propagate upwards in the solar atmosphere and reach the chromospheric layers. We estimate the energy flux carried by a single Alfvén pulse as 1.9–7.7 kW m$^{-2}$, which is more than enough to balance the local upper chromospheric energy losses (of the order of 0.1 kW m$^{-2}$).

## Results

**Chromospheric swirls**. Figure 1b depicts one of 765 high-resolution chromospheric intensity maps close to the disk centre at the Ca II H line core (3968.5 Å) taken at 05:50:01 UT on 5 March 2007 by the SOT. Fifty-five chromospheric swirls have been detected, among which about half (31) rotate in the positive (counter-clockwise, blue) and the other half (24) in the negative direction (clockwise, red). The tracked velocity field and associated swirls determined are shown in more details in the close-up views in Fig. 1c, d. Photospheric swirl detection results from the SOT FG blue continuum (4504.5 Å) observations 118 s earlier are shown in Fig. 1a as a comparison. This time delay is consistent with that we find from the statistical correlation analysis (see below).

In total, 36834 chromospheric swirls have been detected in 764 generated velocity maps, ~49.8% of which rotate in the positive direction, resulting in 48.2 ± 10 swirls in each frame (Fig. 2a). Detections from both Ca II and Hα line core (8542 and 6563 Å) chromospheric observations (see Supplementary Fig. 1) collected at the SST, which sampled a different quiet region and, therefore, provide support of that our results are general, reveal similar numbers (36–44) of swirls in each frame, but within a FOV of ~1600 Mm$^2$ (see Supplementary Figs. 2 and 3). This is consistent with the fact that more photospheric swirls were detected by the SOT observations than by the SST observations[14]. It is likely that the discrepancy in numbers of swirls detected based on the SOT and SST datasets was caused by the fact that SST data have lower cadence and spatial resolution than the SOT data, and also may suffer from residual seeing effects which were not corrected for during data processing. The SOT observations analysed here suggest a swirl population of at least 3.7 × 10$^5$ in the solar chromosphere. The distribution of the effective radius (see definition in Eq. 2) of chromospheric swirls, shown in Fig. 2b, is very similar to that of photospheric swirls, with an average effective radius of ~290 ± 64 km. No significant differences are detected between swirls rotating in positive and negative directions. The average rotating speed (Eq. 2) of chromospheric swirls (~1.8 ± 0.7 km s$^{-1}$, Fig. 2c) is double of the rotating speed of photospheric swirls. The distribution of swirl lifetimes (Fig. 2d) is similar to that of the photospheric swirls[14]. The number of swirls drops nearly exponentially with increasing lifetime, with an average lifetime of ~21 s and maximum likelihood estimation[17] of the exponential rate parameter of ~0.05. More than 60% of chromospheric swirls, initially detected by the ASDA algorithm, have lifetimes less than twice of the cadence (12.8 s), however, swirls observed only in a single frame were discarded from this lifetime analysis. These results do, though, imply that future observations should be performed with higher cadence to determine more accurate properties of swirls.

**Correlation between photospheric and chromospheric swirls**. To study whether the observed photospheric and chromospheric swirls are related, we performed a series of correlation analyses. Corresponding results are shown in Fig. 3. With a time lag of 0 s, we calculated the correlation indices (see the methods section) of all co-temporal photospheric and chromospheric Γ$_2$ (Eq. 1) maps, the average value (CI) of all the correlaton indices and the average percentage of photospheric swirls which overlap with chromospheric swirls. Then, we repeated the above calculation by varying the time lag between photospheric and chromospheric frames from −300 s to 300 s (shifting the photospheric observations from −300 s to 300 s). Corresponding errors were measured by cross-checking the calculation with randomly shuffled datasets 50 times. The blue solid curve in Fig. 3b shows that the CI has a significant peak of ~1%, above the 5σ level (blue shadow), at a time lag of ~125 s. The overlap (black solid curve) reaches a peak of about 41%, above its 5σ level (grey shadow), at a time lag of ~140 s. The same analysis but on a pre-shuffled dataset reveals no

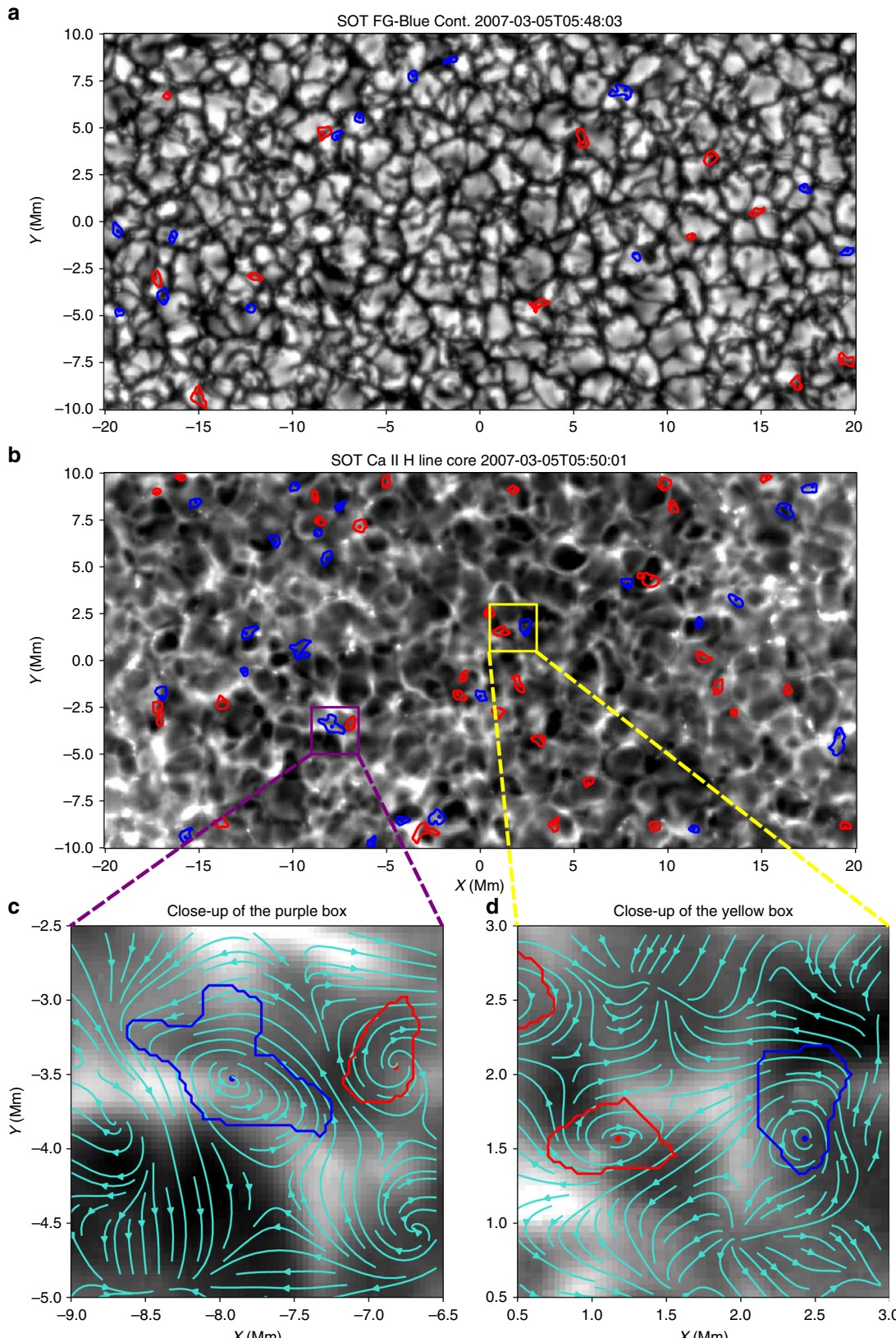

**Fig. 1** Photospheric and chromospheric swirls detected by ASDA using SOT observations. The observed intensity is shown as the white-black background in all panels. Photospheric intensity swirls (**a**) detected at 05:48:03 UT were found to be best correlated with chromospheric swirls (**b**) detected 118 s later (05:50:01 UT). Swirls, with positive (negative) rotating direction are denoted in blue (red). Contours and dots are their edges and centres, respectively. Turquoise arrows in **c**, **d** represent the tracked velocity field by the FLCT method[25,26]. The analysed data contain a series of photospheric and chromospheric observations in a quiet region close to the disk centre. Source data are provided in the Source Data file

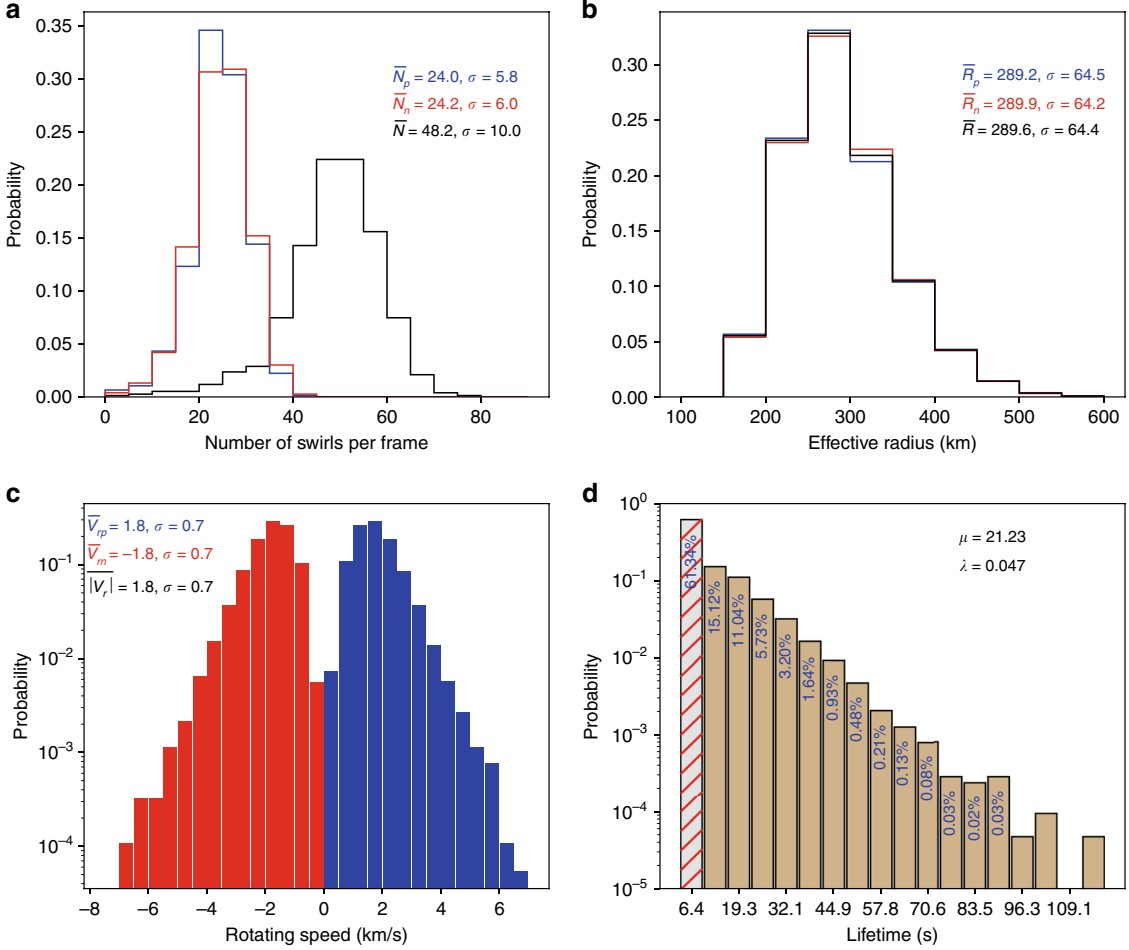

**Fig. 2** Statistics of swirls detected from the SOT Ca II H line core observations. **a** denotes the distribution of number of swirls per frame, with **b** the effective radii, **c** the average rotating speeds at edges and **d** the lifetimes of all swirls. Blue (red) curves, bars and texts in the first three panels represent results of positive (negative) swirls. Black curves and texts are the results of all swirls. $\mu$ and $\lambda$ in **d** are the expectation and maximum likelihood estimation[17] of the exponential rate parameter of the lifetime, respectively. The left most bar is stripped, because lifetimes less than twice of the cadence are not measured but estimated given the limitation on the cadence. These events have also been excluded when estimating $\mu$ and $\lambda$. Source data are provided in the Source Data file

significant peaks above the $3\sigma$ levels at all (see Supplementary Fig. 4a). Similar analysis of the SST observations (see Supplementary Fig. 5) shows that the CI and overlap between Fe I wideband photospheric and Ca II 8542 Å line core chromospheric observations peak at a time lag of 102 and 176 s above the $5\sigma$ levels, respectively. The CI and overlap between the H$\alpha$ 6563 Å line core and Ca II 8542 Å line core chromospheric observations both peak at a time lag close to 0 ($-7.5$ s) above their corresponding $5\sigma$ levels. No peak above $3\sigma$ level has been found when comparing the Fe I wideband photospheric and H$\alpha$ 6563 Å line core chromospheric observations, which might be a result of the extended formation heights of the H$\alpha$ line[18].

A peak overlap of 41% means a high number (suggesting a population of $\sim 0.66 \times 10^5$ expected in the photosphere) of SOT photospheric swirls have been found to correspond to chromospheric swirls. However, it should be remembered that this is the result for just a single time lag between photospheric and chromospheric observations. The inhomogeneity of the magnetic field and plasma properties in the solar atmosphere needs to be kept in mind. Further analysis (see Supplementary Fig. 4b) shows that: about 81% of the detected SOT photospheric swirls (suggesting a population of $\sim 1.30 \times 10^5$ expected in the photosphere) have correspondences in the chromosphere within a time lag range of 100–160 s; and about 94% of the photospheric frames

have more than half swirls overlapping with chromospheric swirls found within a time lag range of 100–160 s. Similarly (see Supplementary Fig. 5d), about 86% of the detected SST Fe I photospheric swirls have correspondences in the Ca II 8542 Å line core chromospheric observations within a time lag range of 100–160 s; and about 99% of the SST photospheric frames have more than half swirls overlapping with chromospheric swirls found within a time lag range of 100–160 s.

The above results suggest that: most of photospheric swirls can be correlated to chromospheric swirls; the mode time-scale required for the swirling motion to propagate from the photosphere to the chromosphere is about 130 s; and, less than 0.01% (Fig. 2) swirls live longer than 100 s, indicating that most swirls fade away before forming large-scale tornadoes and the swirling motion should travel upward as a short ($\sim 20$ s) pulse instead of a wave train. Considering the formation height of the Ca II H line core is between 1000 km and 2000 km[18], the short pulse should travel at a speed of 8–15 km s$^{-1}$.

In order to investigate the nature of the pulses carrying the swirling information, we performed a series of 3D MHD simulations using the Sheffield Advanced Code (SAC)[19] under a realistic gravitationally-stratified atmosphere. At the beginning of the simulation, we introduced, at 450 km above the $\tau_{500} = 1$ level, a rotational motion, with a lifetime of 20 s centred on the

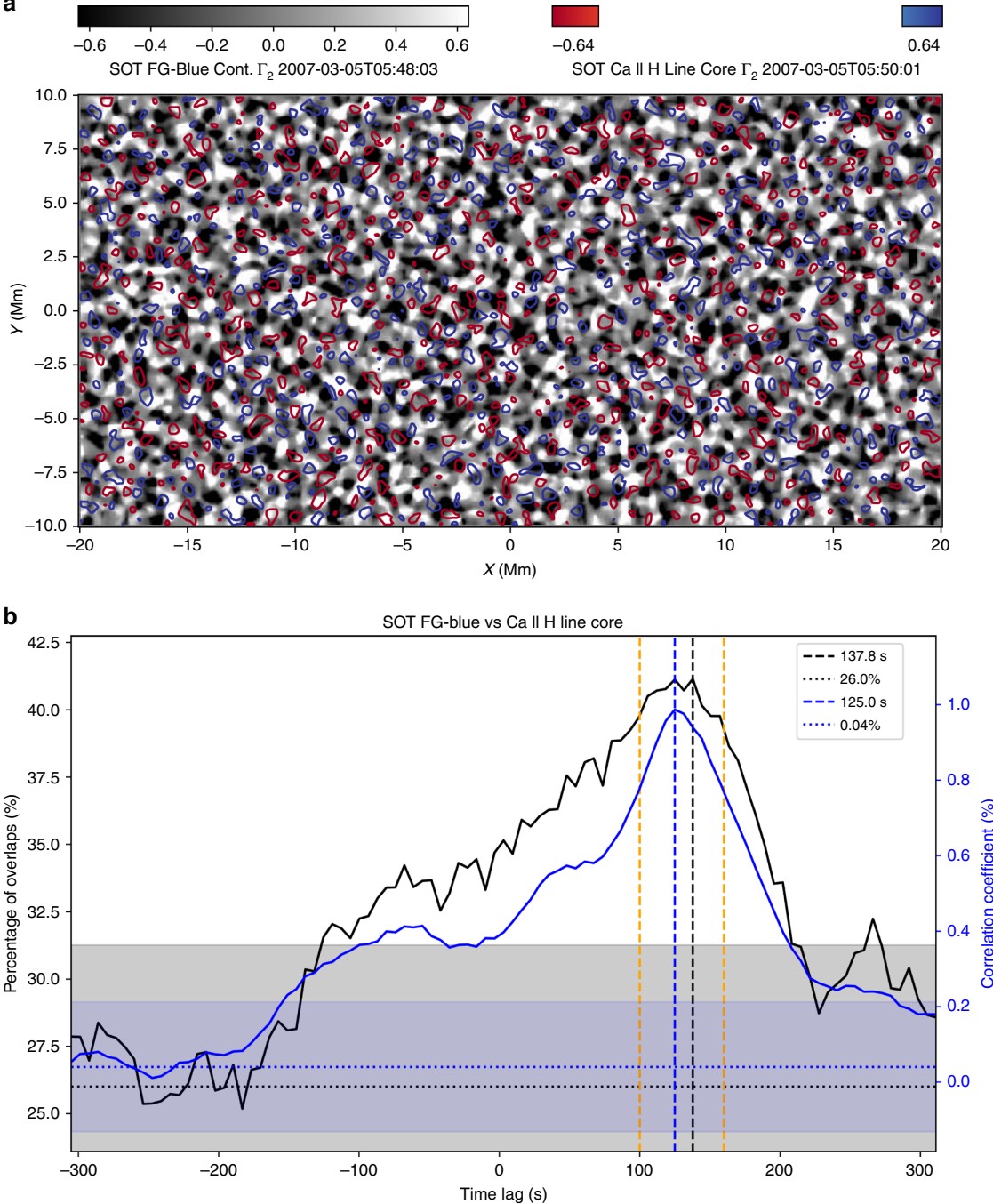

**Fig. 3** Correlations between photospheric and chromospheric swirls detected by the SOT. The black-white background in **a** shows the $\Gamma_2$ distribution of the photospheric observation by the SOT FG-Blue continuum at 05:48:03 UT. It has a peak correlation index (CI ~ 2.4%) with the $\Gamma_2$ distribution of the Ca II H chromospheric observation (red and blue contours) 118 seconds later at 05:50:01. The two unitless colorbars show the scales of $\Gamma_2$. **b** The blue and black solid curves are the CI and percentage of overlaps between the SOT photospheric and chromospheric swirls varying with different time lags, respectively. The vertical blue and black dashed lines represent the peak of the CI and percentage of overlaps at a time lag of 125 and 138 s, respectively. The two vertical orange dashed lines correspond to time lags of 100 and 160 s, respectively. Via performing 50 times of random shuffle on the series of observations, we have also obtained their corresponding levels of errors (blue and black dotted lines, 0.04% and 26%) and $5\sigma$ ranges (blue and grey shadows). Source data are provided in the Source Data file

flux tube. Figure 4a depicts a snapshot of the simulation (see Supplementary Movie 1) at t ≈ 93 s. The perturbation of the rotational motion has now already travelled up to 600 km above the bottom. It is clear from the Supplementary Movie 2 (as well as Fig. 4), that the azimuthal magnetic field perturbation is opposite oriented to the velocity field perturbation, indicating strongly the Alfvén nature of the pulse. The remnant weak velocity and

magnetic field perturbations at the bottom (see Supplementary Movie 2) are suggested to be the trapped initial perturbations due to the very low (~0.5 km s$^{-1}$) local Alfvén speed outside the flux tube (see Supplementary Fig. 6).

In order to further analyse these pulses, a vertical slit was placed close to the centre of the flux tube. Corresponding time-distance plots are shown in Fig. 4b–d. Panel b of the vertical

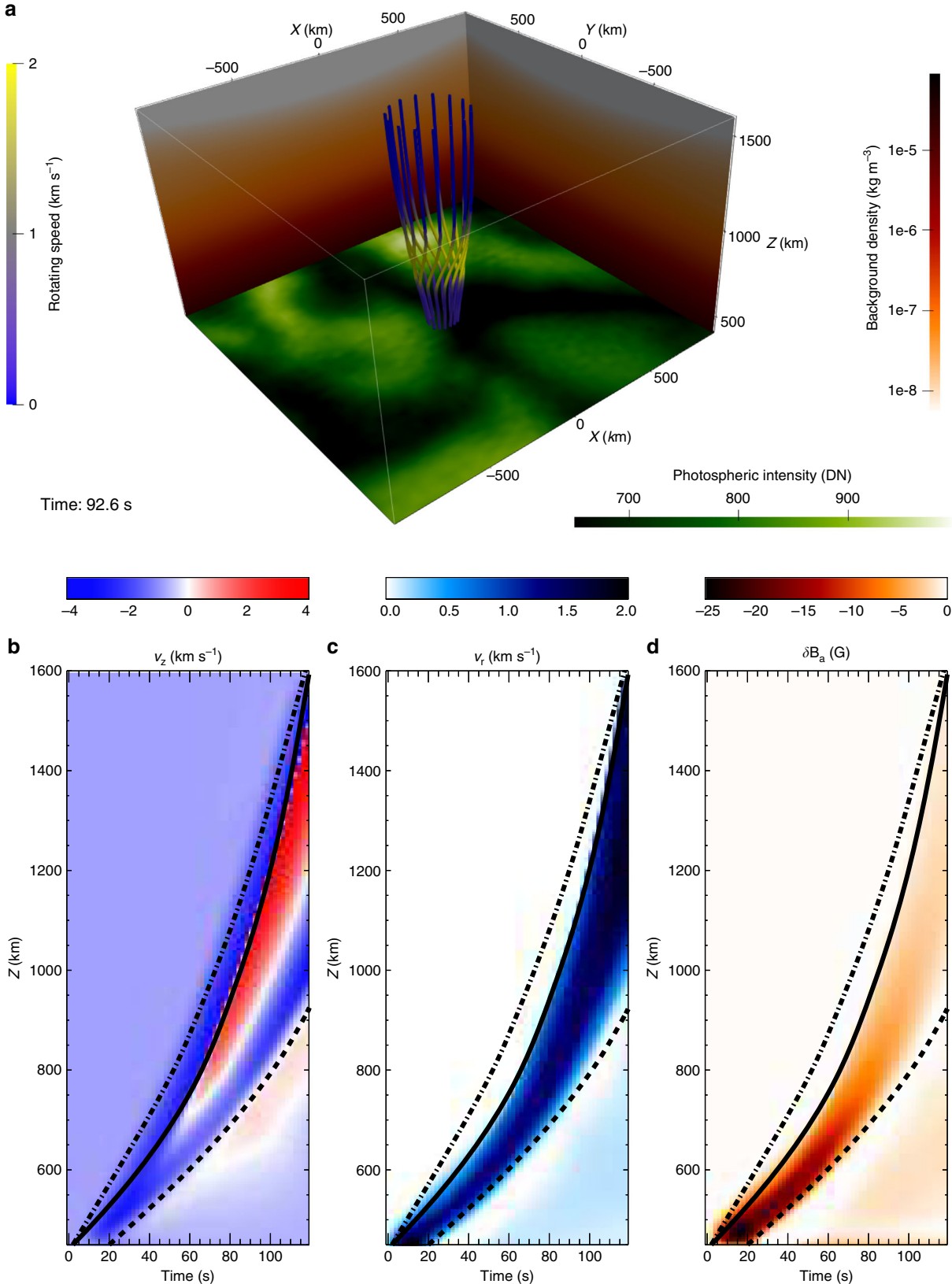

velocity perturbation shows that the pulse travels upward at three different local speeds: the sound speed (dash-dotted curve), the Alfvén speed (solid curve) and the tube speed (dashed curve). However, only the Alfvén pulse carries information of the rotational motion (panel c) and azimuthal magnetic field perturbation (panel d). It takes ~120 s for the Alfvén pulse to travel from the upper photosphere (bottom of the simulation domain) to the upper chromosphere (top of the simulation domain), revealing an average speed of ~9.6 km s$^{-1}$. These simulation results agree well with those on the travelling speed

**Fig. 4** Visualization of the numerical simulation. **a** Snapshot of the simulation at $t \approx 93$ s. Vertical lines are magnetic field lines of the flux tube, with colours denoting the strength of the local rotating speed $v_r$. Red-white vertical backgrounds represent the background plasma density in the simulation. The bottom layer shows part of the SOT FG-blue photospheric intensity observations at 06:03:04 UT when a swirl was detected at the centre of the layer in the intergranular lane. **b**–**d** are the time-distance plots of a perpendicular slit along the z direction located close to the tube centre. $v_z$ and $\delta B_a$ represent the z-direction velocity perturbation and azimuthal magnetic field perturbation, respectively. We find three different pulse fronts in the $v_z$ perturbation, and they travel at local sound (dash-dotted curve), Alfvén (solid curve) and tube (dashed curve) speed, respectively. However, only the Alfvén pulse causes both obvious perturbations in $v_r$ and $\delta B_a$. No wave train can be found because of the short life (20 s) of the driver. Source data of **b**–**d** are provided in the Source Data file

of the swirling motion found from the observations, again confirming the Alfvén-pulse connection between swirls at different heights in the solar atmosphere.

## Discussion

We have found prevalent short-lived photospheric swirls which excite Alfvén pulses, which propagate upward into the upper chromosphere and result in ubiquitous chromospheric swirls (see Supplementary Discussion 1 and Supplementary Movie 3 for further discussions and illustrations). The estimation of the energy flux carried by these Alfvén pulses (see Supplementary Discussion 2) suggests that these abundant events are potentially able to carry a considerable amount of non-thermal energy ($1.9–7.7$ kW m$^{-2}$) into the upper chromosphere. Estimates suggest that this energy flux is enough to balance the local radiative (and other) energy losses[20] (of the order of 0.1 kW m$^{-2}$) in quiet regions. The average energy flux contribution by these Alfvén pulses is estimated to be $33–131$ W m$^{-2}$ (see Supplementary Discussion 2). The exact contribution of these Alfvén pulses to the global upper chromospheric heating needs to be further studied by applying ASDA to observational data with higher spatial and temporal resolutions. Dissipation of these Alfvén pulses has not been accounted for in the simulation, given the limited ability to resolve small scale-instabilities and turbulence. This important topic should be further studied in the future to illustrate how the energy carried by these pulses would be deposited to heat the local upper chromosphere. We shall also note that, in the numerical simulations, the propagation of an Alfvén pulse was studied in an expanding flux tube. Thus, more evidence about the excitation of the Alfvén pulses might be provided, if one could find observationally some (significant) correlation between photospheric swirls and vertical magnetic flux tubes. However, currently available solar magnetic field observations are not suitable for conducting the above study and high-resolution observations or realistic numerical simulations are needed (see Supplementary Discussion 3 and Supplementary Fig. 7 for more details).

Based on the results obtained in this article, we propose future work that should be focused on to discover the complete energy channelling scenario by swirls in the solar atmosphere: given that both swirls and spicules[21,22], are mostly located in inter-granular lanes[14,23], our proposed conjecture is that swirls could also excite spicules through linear (e.g., linear Lorentz force) and/or non-linear (e.g., non-linear Alfvén pulses) processes to supply both momentum and energy into the chromosphere and corona.

## Methods

**Observations**. The SOT data employed consist of blue continuum (FG-blue) images sampled with the wide-band imager with a central wavelength of 4504.5 Å and a band width of 4 Å, and chromospheric images at the Ca II H line core with a central wavelength of 3968.5 Å. The FG-blue data were collected between 05:48:03 UT and 08:29:59 UT on the 5 March 2007, and the Ca II H data between 05:48:06 UT and 07:09:28. Images were targeted at a quiet region centred at $x_c = 5.3''$, $y_c = 4.1''$. The cadence is ~6.42 s. Each of the images has a FOV of ~$56'' \times 28''$ (40.1 Mm × 20.1 Mm), with a image size of $1024 \times 512$ px$^2$

and a pixel size of $0.0545''$ (39.2 km). The level-1 fits files were processed with the fg_prep.pro program available in the SolarSoft IDL packages.

**Swirl Detection Using ASDA**. For every pixel $P$ in a preprocessed, scientifically ready image, two dimensionless parameters[24] are defined as:

$$\Gamma_1(P) = \hat{\mathbf{z}} \cdot \frac{1}{N} \sum_S \frac{\mathbf{n}_{PM} \times \mathbf{v}_M}{|\mathbf{v}_M|},$$
$$\Gamma_2(P) = \hat{\mathbf{z}} \cdot \frac{1}{N} \sum_S \frac{\mathbf{n}_{PM} \times (\mathbf{v}_M - \bar{\mathbf{v}})}{|\mathbf{v}_M - \bar{\mathbf{v}}|}. \qquad (1)$$

Here, $S$ is a two-dimensional region containing the target point $P$, with a size of $N$ pixels. $M$ is a point within $S$. $\mathbf{n}_{PM}$ denotes the normal vector pointing from point $P$ to $M$. $\bar{\mathbf{v}}$ is the average velocity vector within the region $S$ and $\mathbf{v}_M$ is the velocity vector at point $M$. Symbols $| \, |$ and $\times$ are for the mode of vectors and the cross product, respectively. The velocity field is estimated using the Fourier Local Correlation Tracking (FLCT) method[25,26] on successive images from the observations. $\bar{\mathbf{z}}$ denotes the normal vector perpendicular to the observation surface pointing towards the observer.

It has been demonstrated that, $|\Gamma_1|$ peaks at the centre and $|\Gamma_2|$ is larger than $2/\pi$ within the edge of a swirl[24]. To find all swirls in a given frame of observations, the levels of $\pm 2/\pi$ of $\Gamma_2$ are contoured to find out all candidates of swirls; and then candidates with peak $|\Gamma_1|$ values greater than a given threshold (0.89)[14,24] are confirmed as swirls. The given threshold removes all candidates with expanding/shrinking speeds larger than half of their rotating speeds[14]. Detailed tests on a series of synthetic data and realistic numerical simulation data with varying spatial resolutions, have been performed and proved ASDA as a proficient and astute method in accurately detecting solar atmospheric swirls[14].

**Parameters of swirls**. Given the velocity field and edge of a swirl, its effective radius ($R$) and average rotating speed ($v_r$) are determined by:

$$R = \sqrt{\frac{A}{\pi}},$$
$$v_r = \frac{1}{k}\sum_{i=1}^{k} \mathbf{v}_i \cdot \mathbf{n}_i. \qquad (2)$$

Here, $A$ is the area of the swirl. $k$ is the total number of points at the edge of the swirl. $\mathbf{v}_i$ and $\mathbf{n}_i$ are the velocity and the normal vector perpendicular to the local radial direction (from the centre of the swirl) of the $i$th point at the edge, respectively.

Lifetimes of swirls at a single height of observations are estimated following the method proposed in the ASDA paper[14]. Suppose, there are two swirls ($S_1$ and $S_2$) detected in two successive frames. $S_1$ is detected at time $t_0$ and $S_2$ at time $t_0 + \Delta t$, where $\Delta t$ is the cadence of the observation. $S_1$ and $S_2$ will be considered as the same swirl, if:

$$\mathbf{c_1} + \mathbf{v_{c1}} \cdot \Delta t \subset S_2. \qquad (3)$$

Here, $\mathbf{c_1}$ is the location of the centre of swirl $S_1$. $\mathbf{v_{c1}}$ is the speed of the centre of $S_1$. The symbol $\subset$ means belonging to. Taking into account that a swirl may experience changes to its rotational motion through time, we then allow swirls to be missing from one frame when evaluating their lifetimes, i.e., $S_1$ and $S_3$ are still considered as the same swirl if $\mathbf{c_1} + \mathbf{v_{c1}} \cdot 2\Delta t \subset S_3$ is true where $S_3$ is a swirl detected at time $t_0 + 2\Delta t$.

**Estimation of correlation indices and overlaps**. Many difficulties exist in directly comparing swirls at two different layers $L_1$ and $L_2$ (for example $L_1$ in the photosphere and $L_2$ in the chromosphere) to find their overlaps: first of all, swirls confirmed by ASDA are only part of all the candidates. A photospheric candidate might be confirmed as a swirl, but its chromospheric correspondence (if there is any) could be not due to reasons such as too large expanding speed. This case would not be rare because of the common flux tube expansion from the photosphere to the chromosphere. Secondly, magnetic field in the low atmosphere is more or less inclined[27], it is unlikely that a photospheric swirl and its chromospheric correspondence stay at exactly the same horizontal location. And, thirdly the shapes of swirls are found to be irregular, meaning that a photospheric swirl and its chromospheric correspondence are unlikely to be 100% overlapped, even if they stay at exactly the same horizontal location.

To bypass the above difficulties, we have developed the following method to estimate the correlation and overlaps of two given layers ($L_1$ and $L_2$):

Firstly, binarize the $\Gamma_2$ maps of $L_1$ and $L_2$, set all points with values larger than $2/\pi$ as 1 and less than $-2/\pi$ as $-1$. All other points are set to 0. The resulted binarized maps are $\Gamma_2^1$ and $\Gamma_2^2$ for layer $L_1$ and $L_2$, respectively.

Secondly, sum the absolute value of all points in $\Gamma_2^1$ as $T_1$, $\Gamma_2^2$ as $T_2$ respectively. Define $T$ as the smaller one between $T_1$ and $T_2$.

Thirdly, multiply $\Gamma_2^1$ and $\Gamma_2^2$, and obtain the correlation map $C$. Then, we define $(\Sigma C)/T$ as the correlation index (CI) between layers $L_1$ and $L_2$. The above procedures determine that CI can only range between $-1$ to 1, with a higher CI value implying a higher correlation between layers $L_1$ and $L_2$. However, the CI of two given layers cannot give us information of how many swirls are overlapped between layers $L_1$ and $L_2$. For example, a CI of 0.01 could either mean only 1% of the swirls in one layer are overlapped with those in the other layer, or all swirls are overlapped but each swirl has only 1% points overlapped, in extreme cases.

And finally, for a given swirl $S$ detected in layer $L_1$, we map all of its points in to the $C$ map obtained in the previous step and calculate the percentage of points which correspond to positive $C$ values as its own correlation index ($CI_S$). Swirl $S$ is then labelled to have its correspondence (to be overlapped) in layer $L_2$ if $CI > 0$ and $CI_S > t_h$, where $CI$ is the correlation index between layers $L_1$ and $L_2$. $t_h$ is a positive value defined by $\Delta^2/\overline{A}$, where $\Delta$ is the pixel size (39.2 km for the utilized SOT observations) and $\overline{A}$ is the average area of all swirls ($= \pi \overline{R}^2$, $\overline{R}$ is the average effective radius ~290 km), meaning that on average there is at least one point within swirl $S$ corresponding to a positive correlation.

However, we are aware of that the above processes cannot fully overcome all difficulties raised above, especially the influence of the inclined magnetic fields. We suggest that the number of photospheric swirls which were found to have their correspondences in the chromosphere using the above method should have been under-estimated.

**Numerical simulations**. The 3D MHD simulations have been performed using the Sheffield Advanced Code (SAC)[19], which solves the ideal MHD equations with the presence of arbitrary perturbations in a gravitationally stratified and magnetised atmosphere. SAC separates background and perturbation variables in order to accurately resolve perturbations in a stratified atmosphere. This approach has been developed with the capability to perform simulations even in the non-linear regime. SAC has been tested and used to study wave propagations along a flux tube previously[10]. The governing equations of SAC are briefly recapped in the Supplementary Note 2.

The initial density and temperature profiles of the simulation performed in this work have been constructed using the VAL IIIC model[28]. An axisymmetric and self-similar expanding flux tube, with a magnetic field strength of 800 G at its footpoint constructed following previous literatures[10,29], is then embedded into the ambient atmosphere. The analytic equations and parameters used for the construction of the magnetic flux tube are available in the Supplementary Note 2. The computational domain of the particular simulation is $-1.0 \leq x \leq 1.0$, $-1.0 \leq y \leq 1.0$ and $0.4 \leq z \leq 1.6$ Mm, simulating from the upper photosphere to the upper chromosphere. The above domain is resolved with a grid size of (129, 129, 259) points in the $x-$, $y-$ and $z$-direction, respectively. Thus the spatial resolutions in the $x-$, $y-$ and $z$-direction are 15.5, 15.5 and 4.6 km, respectively.

The rotational driver is introduced at the bottom of the flux tube at the simulation time $t = 0$, generated from the following formula[10]:

$$v_r = v_0 \cdot \exp\left(-\frac{r^2}{\delta r^2}\right) \cdot \exp\left[-\frac{(z-z_0)^2}{\delta z^2}\right] \cdot \sin\left(\frac{\pi t}{P}\right). \qquad (4)$$

Here, $v_r$ is the rotational speed. $v_0$ is the amplitude and set to be $1$ km s$^{-1}$ according to the observations. $r$ and $z$ are the distance to the centre of the flux tube and height of a point $(x, y, z)$ in the computational domain. $\delta r$ equals to 300 km, corresponding to the average effective radius of observed swirls. $z_0 = 450$ km and $\delta z = 25$ km is the vertical centre and expansion of the driver. $t$ is the simulation time and $P = 20$ s is the lifetime of the driver. $v_0$ is set to 0 after $t = 20$ s. The above setup allows the amplitude of the rotational driver to gradually increase to its maximum before $t = 10$ s and decrease to 0 at $t = 20$ s.

## Data availability
Raw data of the SOT observations are available at http://sot.lmsal.com/. Observational data from the SST and derived data supporting the findings of this study are available from the corresponding author upon reasonable request. The source data underlying all figures are provided as a Source Data file. Each figure corresponds to one npz file (except that Supplementary Fig. 5 corresponds to a zip file that consists of four npz files for the four panels) in the source data file. These npz files can be restored using the NumPy package in Python. Each npz file contains a variable named as readme providing the explanations of every variable stored.

## Code availability
The source code of ASDA written in Python is open-access, available at https://github.com/PyDL/asda. The Python wrapper for the FLCT is available at https://github.com/PyDL/pyflct. The source code of SAC is available at http://ascl.net/1306.001 and https://github.com/SWAT-Sheffield/SAC. The SolarSoft IDL package is available at https://sohowww.nascom.nasa.gov/solarsoft/. The NumPy package for Python is available at https://www.numpy.org/. Codes used for the correlation analysis are available upon reasonable request.

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

## Acknowledgements

We thank the Science and Technology Facilities Council (STFC, grant numbers ST/M000826/1, ST/L006316/1) for the support to conduct this research. CJN also acknowledges support by STFC consolidated grant ST/P000304/1. YW acknowledges support by NSFC 41774178, 41574165 and 41761134088. Hinode is a Japanese mission developed and launched by ISAS/JAXA, with NAOJ as domestic partner and NASA and STFC (UK) as international partners. Hinode is operated by these agencies in co-operation with ESA and NSC (Norway). The Swedish 1-m Solar Telescope is operated on the island of La Palma by the Institute for Solar Physics of Stockholm University in the Spanish Observatorio del Roque de los Muchachos of the Instituto de Astrofísica de Canarias. The Institute for Solar Physics is supported by a grant for research infra-structures of national importance from the Swedish Research Council (registration number 2017-00625). The SST data was collected as part of the observing time proposal awarded in 2012 to RE as the PI. We thank the SOLARNET for the awarding of the observing time and the data reduction.

## Author contributions

J.L. led the development of the code. J.L. and C.J.N. performed the analysis of the data. J.L. conducted the numerical simulation, with initial advices from B.S., Y.W. and R.E. led the interpretation of the results. R.E. was the P.I. of the ground-based observations and has led the overall research. All authors reviewed the paper.

## Additional information

**Competing interests:** The authors declare no competing interests.

**Peer Review Information**: Nature Communications thanks the anonymous reviewers for their contribution to the peer review of this work. Peer reviewer reports are available.

