## [Peer Review File · Nature Communications]

Reviewers' comments:

Reviewer #1 (Remarks to the Author):

Report on Manuscript #NCOMMS-18-26662

Title: Ubiquitous Alfvén pulses supply enough energy to heat the upper solar chromosphere

Authors: J. Liu, C. J. Nelson, B. Snow, Y. Wang, R. Erdélyi

The authors claim that short-lived swirls observed in the solar photosphere excite Alfvén pulses, which propagate upwards into the upper solar chromosphere. These pulses would then be able to transport sufficient energy to balance energy loss in quiet Sun regions, thus suggesting a viable mechanism to heat the quiet regions in the entire upper chromosphere.

The topic of the manuscript is certainly of interest and importance. To trace and characterise the propagation of energy through the different layers of the solar atmosphere is essential to find a solution to the chromospheric and coronal heating problems. The authors employ state-of-the-art instrumentation with data of the highest available quality. A multi wavelength multi instrument analysis is performed. Also, advanced numerical computations were carried out trying to mimic the generation and propagation of the disturbances thought to be responsible for the observed swirl propagation.

The study shows some unquestionable pieces of evidence:

1) Observations by Hinode/SOT and SST/CRISP show the presence of a large sample of

photospheric and chromospheric swirls in a large sample. The statistical properties of these populations are analysed.

2) By means of correlation and time-lag analyses, the authors find co-spatial and co-temporal relationships between both samples. Correlation is found in about 40% of the cases. The chromospheric swirls would therefore be caused by the upward propagation of the photospheric ones. The time it takes to the swirls to travel upwards is statistically calculated to be around 130 sec.

3) Support to this idea was sought by performing numerical simulations of the excitation and propagation of Alfvén pulses. The hypothesis is made that rotational motions introduced above the $\tau_{500}=1$ level can propagate upward in the form of Alfvén pulses. The time it takes them to travel from photosphere to chromosphere agrees well with the observed estimate.

4) Analytically calculated energy estimates are given, which seem to point to a large reservoir of energy flux.

These results are certainly important since they provide convincing evidence about the possible importance of this numerous small-scales dynamics on the photosphere-chromosphere energetics. Observational evidence about swirls in the solar photosphere is known since decades ago.

The merit of this study is in observationally finding their chromospheric counterparts and in providing statistical evidence about their relationship in space and time.

However, these findings do not support the strong claims being made in the title of the manuscript and in the concluding paragraph of the manuscript. It is said that these pulse supply enough energy to

heat the chromosphere. But the following concerns immediately arise:

The average energy flux carried out by the Alfvén pulses is calculated using simple analytical relationship for the energy flux of plane Alfvén waves in an infinite and uniform medium and by inserting values for the physical parameters in the simulation.

This energy of the pulses could have been computed from the numerical simulations, but these simulations

do not even model the process of dissipation of energy nor the heating associated to this dissipation.

To conclude that any heating can arise from these simulated pulses is therefore an overstatement, because

the numerics does not model the dissipation of energy, not the heating associated to this dissipation.

What is the heating mechanism? Also, why do I see rotational motions in the driver, but not thereafter

when the disturbance propagate upwards?

The authors show observational evidence for the presence of swirls but do not offer convincing evidence about them to be important for chromospheric heating. No dissipation mechanism is discussed. No energy calculation is performed in which one could see how much of the energy can be dissipated. No series of interconnected events are suggested for the full process that is being invoked. For instance, in the concluding paragraph, the authors admit that "future work should be focused on to discover the complete energy channeling scenario by swirls in the solar atmosphere". It is precisely this type of evidence that I was expecting this paper to show to merit publication in Nature Communications.

For this reasons, I do not recommend acceptance.

Reviewer #2 (Remarks to the Author):

Referee report of the paper on

Ubiquitous Alfvén Pulses Supply Enough Energy to Heat the Upper Solar Chromosphere

By Liu, J., Nelson, C. J., Snow, B., Wang, Y., Erdelyi, R.

Submitted for publication in Nature Communication (2018 Sep)

This is an interesting paper on the long standing puzzle, the heating mechanism of the solar chromosphere and corona. The idea to compare swirls in the photosphere and those in the chromosphere is very interesting and good. However, there are some fundamental questions on the method and results. Hence I cannot recommend publication at the present stage. I would like to encourage further study of the problem with this approach and revise the paper. My questions and comments are as follows:

Major Points:

(1) The authors did not show the relation between the swirls and the small scale vertical magnetic flux tubes. However, in order to excite the Alfvén pulse along the vertical flux tube, photospheric swirls must be present just on the small scale vertical magnetic flux tubes in the photosphere. Please study this correlation, and please show such results on the correlation between swirls and flux tubes in the revised paper.

(2) How much Poynting energy flux is carried by the Alfvén pulse into the chromosphere from the photosphere ?

Minor points:

(3) According to the results, there are 55 swirls in the chromosphere, while there are 21 swirls in the photosphere. Why there are more swirls in the chromosphere than in the photosphere ?

(4) In the 4th paragraph of page 2, the author wrote “the azimuthal magnetic field perturbation is opposite oriented to the velocity field perturbation”. I cannot understand this sentence. How can I understand the direction of azimuthal velocity and magnetic field vector from Figure 4 ?

(5) Figure 1: Please show the images of photospheric swirls in the photosphere

(6) Figure 4: Is ΔB_h the same as ΔB_a ?

(7) Reference 19 (De Pontiou et al. 2004) is not the paper on Alfvén waves nor Alfvénic waves. Instead, the author should cite Kudoh and Shibata (1999) ApJ 514, 494 which is an important paper on the Alfvén wave model of spicules and chromospheric/coronal heating.

Reviewer #3 (Remarks to the Author):

The paper aims at providing observational evidence that Alfvén wave pulses excited by swirls in the solar photosphere are responsible for chromospheric and coronal heating. The authors perform correlation analysis between swirls detected at different heights in the solar atmosphere, and provide results of 3D MHD numerical simulation of these dynamical structures. However, the statistical analysis clearly shows that the time resolution of the study is inadequate, since most swirls are not well resolved, and that the correlation between the photospheric and chromospheric swirls is weak. The authors suggest that the Alfvén pulses are prevalent, and that they propagate upwards through the solar atmosphere and reach the chromospheric layers. The authors claim the average energy flux carried by the Alfvén pulses associated with the swirls are $33 - 131 \text{ W m}^{-2}$, and suggests that these events could transport a considerable amount of non-thermal energy into the upper chromosphere to balance the radiative (and other) energy loss. Overall, the radiative energy losses in the chromosphere are two order of magnitude larger than the estimates energy flux provided by the swirls in this study. While the upper chromosphere has lower radiative losses than the mid-chromosphere, there is no evidence that the presumed Alfvénic pulses would preferentially deposit their energy in the upper chromosphere. While it is plausible that Alfvén waves can heat the corona, this is not adequately supported by the present analysis. In summary the paper does not provide convincing evidence that swirls and the associated waves that may be produced, can supply adequate energy flux for chromospheric heating, and the statistical analysis is weak. Therefore, I cannot recommend publication for this study.

Minor comments:

The title implies very firm conclusion of the study. However, the analysis in the paper does not support such firm conclusion. Please revise.

There are minor English grammar issues through the paper that need attention. For example, in the abstract: "in stead" => "instead"; "the enough" => "enough", etc.

Please clarify “when assessing the affinity to dissipate these perturbations.”

Lines 80-88 discuss 3D MHD modeling of the swirls. However, additional details of the modeling, such as adequacy of the resolution, numerical dissipation effects, and other simplifications and assumptions of the model should be given in the paper or in the supplement.

The provided numerical simulation results are not entirely new, already discussed in detail by Fedun et al. (2011).

Response to Reviewers' Comments

Reviewer #1 (Remarks to the Author):

The authors claim that short-lived swirls observed in the solar photosphere excite Alfvén pulses, which propagate upwards into the upper solar chromosphere. These pulses would then be able to transport sufficient energy to balance energy loss in quiet Sun regions, thus suggesting a viable mechanism to heat the quiet regions in the entire upper chromosphere.

The topic of the manuscript is certainly of interest and importance. To trace and characterise the propagation of energy through the different layers of the solar atmosphere is essential to find a solution to the chromospheric and coronal heating problems. The authors employ state-of-the-art instrumentation with data of the highest available quality. A multi wavelength multi instrument analysis is performed. Also, advanced numerical computations were carried out trying to mimic the generation and propagation of the disturbances thought to be responsible for the observed swirl propagation.

The study shows some unquestionable pieces of evidence:

1) Observations by Hinode/SOT and SST/CRISP show the presence of a large sample of photospheric and chromospheric swirls in a large sample. The statistical properties of these populations are analysed.

2) By means of correlation and time-lag analyses, the authors find co-spatial and co-temporal relationships between both samples. Correlation is found in about 40% of the cases. The chromospheric swirls would therefore be caused by the upward propagation of the photospheric ones. The time it takes to the swirls to travel upwards is statistically calculated to be around 130 sec.

3) Support to this idea was sought by performing numerical simulations of the excitation and propagation of Alfvén pulses. The hypothesis is made that rotational motions introduced above the $\tau_{500=1}$ level can propagate upward in the form of Alfvén pulses. The time it takes them to travel from photosphere to chromosphere agrees well with the observed estimate.

4) Analytically calculated energy estimates are given, which seem to point to a large reservoir of energy flux.

These results are certainly important since they provide convincing evidence about the possible importance of this numerous small-scales dynamics on the photosphere-chromosphere energetics. Observational evidence about swirls in the solar photosphere is known since decades ago. The merit of this study is in observationally finding their chromospheric counterparts and in providing statistical evidence about their relationship in space and time.

Response 1.1: We appreciate that the reviewer has thought very highly of our paper and summarized some of our important innovations.

However, these findings do not support the strong claims being made in the title of the manuscript and in the concluding paragraph of the manuscript. It is said that these pulse supply enough energy to heat the chromosphere. But the following concerns immediately arise: The average energy flux carried out by the Alfvén pulses is calculated using simple analytical relationship for the energy flux of plane Alfvén waves in an infinite and uniform medium and by inserting values for the physical parameters in the simulation.

This energy of the pulses could have been computed from the numerical simulations, but these simulations do not even model the process of dissipation of energy nor the heating associated to this dissipation.

To conclude that any heating can arise from these simulated pulses is therefore an overstatement,

because the numerics does not model the dissipation of energy, not the heating associated to this dissipation. What is the heating mechanism? Also, why do I see rotational motions in the driver, but not thereafter when the disturbance propagate upwards?

The authors show observational evidence for the presence of swirls but do not offer convincing evidence about them to be important for chromospheric heating. No dissipation mechanism is discussed. No energy calculation is performed in which one could see how much of the energy can be dissipated. No series of interconnected events are suggested for the full process that is being invoked. For instance, in the concluding paragraph, the authors admit that "future work should be focused on to discover the complete energy channeling scenario by swirls in the solar atmosphere". It is precisely this type of evidence that I was expecting this paper to show to merit publication in Nature Communications.

Response 1.2: We agree that the dissipation mechanism is indeed a very important part in solving the heating problem. However, we would like to emphasise that it is also a completely different topic from what we are currently aiming to address in the paper. As far as we are aware of, the damping mechanism of MHD waves has been majorly underpinned by energy cascade from large to small scales. There are few different theories trying to explain the dissipation mechanism of MHD waves in the solar atmosphere: 1) mode coupling (e.g., Pascoe et al. 2013); and 2) instability or turbulence (e.g., Parker 1991; Oughton et al. 2001; Cranmer & van Ballegoijen 2005; Verdini et al. 2010, just to name a few from the vast literature on this topic). However:

- 1) There is still an intense debate about the applicability of these different heating mechanisms to dissipate the transported energy effectively.
- 2) Most papers on the heating mechanisms have been focused on theories or numerical simulations. Until now, *no solid direct and decisive observations* have been provided to the community of which mechanism is ubiquitous and is playing the most important role in dissipating wave energy in the solar atmosphere.
- 3) The size of swirls we have found in our paper is already at the limit of current resolution, and it is impossible to directly **observe** the cascade to even smaller scales with the current facilities.

Based on the above facts and indeed that our supporting simulation was not designed for tackling the dissipation problem, we would not wish to make any **conjecture or speculations** on the dissipation mechanism of the observed Alfvén pulses. We do not know the exact heating mechanism yet, any conjecture without solid support from direct observations may mislead the community and affect the scientific reliability of our paper.

We would like to stress that, the purpose of this paper, is to **provide direct and solid evidence of the presence of Alfvén pulses**, which could carry a considerable amount of energy and have potential abilities to heat the quiet upper chromosphere, propagating from the photosphere to the chromosphere. **We, here, did not intend to solve the other problem of how the waves/pulses would dissipate their energy in the solar atmosphere, just as many other observations (to name a few, Wedemeyer et al. Nature 2012; Martínez-Sykora et al. Science 2017) did not tackle this problem either. Dissipation is a related but completely different problem.**

To avoid misunderstanding, we have now revised the title, as well as the text in the conclusion to make our intention clearer (see text in purple in the revised version).

why do I see rotational motions in the driver, but not thereafter when the disturbance propagate upwards?

Response 1.3: We are afraid that the referee might have misunderstood the nature of these Alfvén pulses. The rotational motion in the driver, at the bottom of the simulation, shows how the initial perturbation is introduced into the flux tube. Of course, one will not see any rotation in the driver after

the driver stops. And, also, one will neither see the magnetic field “rotating” after the driver stops. Instead, we see the perturbation (i.e. “twist”) of the magnetic field propagates upward at the local Alfvén speed. However, as you can see from Movie 3, when the magnetic field perturbation passes by a surface, you can find the rotational motion of plasma if it is not homogeneous.

Reviewer #2 (Remarks to the Author):

This is an interesting paper on the long standing puzzle, the heating mechanism of the solar chromosphere and corona. The idea to compare swirls in the photosphere and those in the chromosphere is very interesting and good. However, there are some fundamental questions on the method and results. Hence I cannot recommend publication at the present stage. I would like to encourage further study of the problem with this approach and revise the paper. My questions and comments are as follows:

Major Points:

(1) The authors did not show the relation between the swirls and the small scale vertical magnetic flux tubes. However, in order to excite the Alfvén pulse along the vertical flux tube, photospheric swirls must be present just on the small scale vertical magnetic flux tubes in the photosphere. Please study this correlation, and please show such results on the correlation between swirls and flux tubes in the revised paper.

Response 2.1: We agree that, this is certainly a very interesting topic. If we see some significant correlation between photospheric swirls and small-scale vertical magnetic flux tubes, we will have more evidence about the excitation of the Alfvén pulses. **However, what are the observational signatures of magnetic flux tubes?** We could be almost sure that, magnetic bright points (MBPs, e.g., Almeida et al. 2010) represent small-scale magnetic flux tubes with strong (up to kG) magnetic field. **But, are there small-scale magnetic flux tubes if there is no MBP? We are afraid that the answer is yes!**

Even if we ignore the above dilemma, we have found it is impossible to study the correlation between the detected swirls and MBPs using the currently available data because:

1) SST observations could provide observations of MBPs with barely enough resolution. However, unfortunately, we do not have enough Stokes observations to derive magnetic field information in our current available dataset, even if we ignore the influence of the seeing effect on the observation of MBPs.

2) HMI data might be a candidate, however, its resolution (with a pixel size of more than 440 km, ~10 times of that of the utilized SST observations) seems too large. Given that the typical radius of a photospheric MBP is around 100 km (Keys et al. 2013), its magnetic field would be averaged to as low as 30 to 60 G in HMI observations, if its original magnetic field strength is, say, a typical value of 500 to 1000 G. It means, if we see a bright pixel with magnetic field just above 3 times of the observational error (10 G, Liu et al. 2012), we may have no idea whether it is an “averaged” MBP, or a weak region with no MBP, or just simply within the 3σ error.

Ignoring the above effects for a moment, and following the spirit of your suggestion in your report, we have done a series of tests using HMI LOS magnetograms. Figure 1 in this response below shows an example of the absolute LOS magnetic field with a scale from 0 (white) to 50 G (black), in the FOV of the corresponding SST observations. HMI data has been aligned to match the location and orientation of SST observations, using information derived from feature comparison between SST 6302 Å wideband and AIA 1700 Å observations. We can see how coarse the HMI observation is and how small-scale magnetic elements are smoothed over pixels.

Figure 1. Background: HMI LOS magnetogram at 08:07:26 UT in the FOV of the studied SST observations in the paper. Red and blue contours: photospheric intensity swirl detected from SST 6302 Å wideband observations at all most the same time.

Red and blue contours are the detected SST photospheric intensity swirls. Now, we use an *extremely loose criterion*: any swirl with even one point having absolute magnetic field strength above 30 G is considered to have a strong magnetic field (and thus marked as corresponding to a magnetic flux tube), and is contoured out with solid lines. All others are contoured out as dashed lines. As we can see, there is only 1 swirl corresponding to strong magnetic field in the HMI image.

We have further studied all 77 frames of the HMI observations during the period of the utilized SST observations in the paper, by exploring their correlation with photospheric intensity swirls detected in their closest (in time) SST frames. It turns out that, only 3.3% of the swirls have been found to correspond to strong HMI magnetic field regions. As a comparison, we further did a Monte-Carlo test by comparing each HMI observation with random SST swirl detection result. *Unsurprisingly*, again, 3.3% swirls have been found to correspond to strong HMI magnetic field regions.

All the above results indicate that, *HMI observations are simply too coarse* to be used in this study and we would prefer not to include any of the above tests in our present paper.

To conclude, we are not able to see any reliable solution for this particular problem using solar

observations at the *current* stage. As far as we are aware, there are possibly two ways to study this particular problem (both of which would be well outside the current scope of the article, which is to show the evidence of these pulses propagating upwards through the atmosphere): 1) applying swirl detection and MBP detection algorithms to realistic simulation data (for example Bifrost). This will be one of our future avenues of work, however, it would require access to Bifrost simulation data and the development of (or use of if there is openly available) an automated MBP detection algorithm; 2) using the high-resolution magnetic field observations from DKIST would be another good choice. A science user case, under development and called UC178 on the DKIST CSP list, will be submitted to the DKIST project on this particular topic in the future.

(2) How much Poynting energy flux is carried by the Alfvén pulse into the chromosphere from the photosphere?

Response 2.2: Using Eq. S3 in the Supplementary material, $\rho=1.5\times 10^{-5}$ kg m⁻³ from the simulation at the bottom of the simulation ($z=450$ km), $v\approx 1.0$ km s⁻¹ from photospheric swirl detection and $c_A=0.4$ km s⁻¹ at $z=450$ km in the simulation, we estimate that the energy flux carried by a single Alfvén pulse from the photosphere is about 6.0 kW m⁻². We shall also note that, according to Verma et al. 2013, local correlation tracking was found to underestimate the photospheric horizontal velocity field by a factor of as much as 3, the above estimated energy flux should be the lower limit.

Minor points:

(3) According to the results, there are 55 swirls in the chromosphere, while there are 21 swirls in the photosphere. Why there are more swirls in the chromosphere than in the photosphere ?

Response 2.3: There could be multiple causes for this, including but not limited to: 1) photospheric intensity is an integration over different heights in the photosphere, meaning more noise; 2) rotating speed of photospheric intensity swirls is rather small (half of that of chromospheric swirls), as we can see from the results, meaning that many of them would not be resolved by the combination of the FLCT (which already usually underestimate the photospheric velocity by a factor of as much as three, Verma et al. 2013) and swirl detection algorithm (which highly relies on the rotational speed); 3) there is different density inhomogeneity in the photosphere and chromosphere, meaning some swirls would not be detected if the local plasma is not inhomogeneous enough. Exact reasons need to be confirmed using simulation data, however, this is beyond the present scope of the paper.

The above discussion has now been added to the Supplementary material in the revised version.

(4) In the 4th paragraph of page 2, the author wrote “the azimuthal magnetic field perturbation is opposite oriented to the velocity field perturbation”. I cannot understand this sentence. How can I understand the direction of azimuthal velocity and magnetic field vector from Figure 4 ?

Response 2.4: From Movie M2 and Figure 4, we can both see the opposite orientation between the azimuthal magnetic field perturbation and velocity field perturbation. In Movie M2, positive v_x corresponds to negative δB_x . In Figure 4, v_r is positive while δB_a is negative.

(5) Figure 1: Please show the images of photospheric swirls in the photosphere

Response 2.5: Now added.

(6) Figure 4: Is δB_h the same as δB_a ?

Response 2.6: Yes, now corrected to δB_a .

(7) Reference 19 (De Pontiou et al. 2004) is not the paper on Alfvén waves nor Alfvénic waves. Instead, the author should cite Kudoh and Shibata (1999) ApJ 514, 494 which is an important paper on the Alfvén wave model of spicules and chromospheric/coronal heating.

Response 2.7: De Pontieu et al. 2004 (reference 20) is cited because of spicules, not because of Alfvén waves. Now, we have also added Kudoh and Shibata (1999) in the reference before De Pontieu et al. 2004.

Reviewer #3 (Remarks to the Author):

The paper aims at providing observational evidence that Alfvén wave pulses excited by swirls in the solar photosphere are responsible for chromospheric and coronal heating. The authors perform correlation analysis between swirls detected at different heights in the solar atmosphere, and provide results of 3D MHD numerical simulation of these dynamical structures. However, the statistical analysis clearly shows that the time resolution of the study is inadequate, since most swirls are not well resolved, and that the correlation between the photospheric and chromospheric swirls is weak.

Response 3.1: Thanks for pointing out one of the reasons why the CI is relatively low. However, after comparing to the noise analysis, we can find both the peak CI and peak percentage of overlaps are *well above 5 σ errors*, indicating strong (not weak) peaks at the given time lag.

The authors suggest that the Alfvén pulses are prevalent, and that they propagate upwards through the solar atmosphere and reach the chromospheric layers. The authors claim the average energy flux carried by the Alfvén pulses associated with the swirls are 33 - 131 W m⁻², and suggests that these events could transport a considerable amount of non-thermal energy into the upper chromosphere to balance the radiative (and other) energy loss. Overall, the radiative energy losses in the chromosphere are two order of magnitude larger than the estimates energy flux provided by the swirls in this study.

Response 3.2: It is possible that the referee might have mixed up the radiative energy losses in the **quiet** chromosphere with the **active** chromosphere. According to Table 10.1 in “Magnetohydrodynamics of the Sun” by E. Priest (and references therein), the radiation loss in the quiet and active upper chromosphere is 300 W m⁻² and 2000 W m⁻², respectively. These values for quiet Sun regions (100 to 300 W m⁻²) have also been widely used in the literature (e.g., Priest et al. Nature 1998; Schrijver et al. Nature 1998; De Pontieu et al. Science 2007; Cirtain et al. Science 2007; McIntosh et al. Nature 2011; Wedemeyer-Bohm et al. Nature 2012; Martínez-Sykora et al. Science 2017) and are in line with our energy estimates.

While the upper chromosphere has lower radiative losses than the mid-chromosphere, there is no evidence that the presumed Alfvénic pulses would preferentially deposit their energy in the upper chromosphere.

Response 3.3: As discussed in our response to referee 1, we do not know where the Alfvén pulses prefer to deposit their energy, because our simulation was not designed to treat such small scales where the turbulence happens. But, as we can see from our calculation, at least 1.7 to 7.7 kW m⁻² of energy does arrive successfully in the upper chromosphere for any single Alfvén pulse. By the way, we would prefer not to call them “Alfvénic”, which is an ambiguous concept mixing up Alfvén and kink modes.

While it is plausible that Alfvén waves can heat the corona, this is not adequately supported by the present analysis. In summary the paper does not provide convincing evidence that swirls and the associated waves that may be produced, can supply adequate energy flux for chromospheric heating, and the statistical analysis is weak. Therefore, I cannot recommend publication for this study.

Response 3.4: In our conclusion, we only comment on the potential of these Alfvén pulses in heating the upper chromosphere, *not the corona*. We do not have statement like “we have adequate evidence of these Alfvén pulses to heat the corona” in our paper.

Minor comments:

The title implies very firm conclusion of the study. However, the analysis in the paper does not support such firm conclusion. Please revise.

Response 3.5: Title and corresponding text have been revised.

There are minor English grammar issues through the paper that need attention. For example, in the abstract: “in stead”=> “instead”; “the enough” => “enough”, etc.

Response 3.6: Corrected.

Please clarify “when assessing the affinity to dissipate these perturbations.”

Response 3.7: Now, we have added, “because we do not yet know whether pulses dissipate in a different way than waves” before this sentence.

Lines 80-88 discuss 3D MHD modeling of the swirls. However, additional details of the modeling, such as adequacy of the resolution, numerical dissipation effects, and other simplifications and assumptions of the model should be given in the paper or in the supplement.

The provided numerical simulation results are not entirely new, already discussed in detail by Fedun et al. (2011).

Response 3.8: The above two comments conflict with each other. Because the simulation was adapted from Fedun et al. 2011, we cited this work 3 times in our manuscript. And, we never stated that the simulation results are entirely new. Again, because Fedun et al. 2011 has already addressed enough details of the simulation, there is no need to repeat them in our paper given that we have properly cited it (i.e., 3 times).

Reviewers' comments:

Reviewer #1 (Remarks to the Author):

Report on Manuscript #NCOMMS-18-26662A-Z

Title: Evidence of ubiquitous Alfvén pulses transporting energy from the photosphere to the upper chromosphere

Authors: J. Liu, C. J. Nelson, B. Snow, Y. Wang, R. Erdélyi

This is a revised version of a previously submitted manuscript. The original manuscript contained strong claims about propagating Alfvén pulses found to carry enough energy to sustain the high temperatures of quiet regions in the entire upper atmosphere. These claims were not supported by the presented evidence.

The authors have now lessened their claims and present a fair manuscript where the finding and properties of photospheric and chromospheric short-lived swirls and their statistical relationship are presented.

An estimate about the energy potentially to be carried by these events is presented. The magnitude is high enough for these events to be considered important in the energy transport across the solar atmosphere.

The authors employ state-of-the-art instrumentation with data of the highest available quality. A multi wavelength multi instrument analysis is performed. Also, advanced numerical computations were carried out trying to mimic the generation and propagation of the disturbances thought to be responsible for the observed swirl propagation.

I find these results are certainly of interest to the solar community and perhaps also to readers from other fields. The claims are now in accordance with the level of evidence being presented. For these reasons, I recommend the manuscript for acceptance in Nature Communications.

Reviewer #2 (Remarks to the Author):

The authors responded various comments and questions by the referee seriously to some extent, which improved the paper very much. Only one concern is the relation between the swirls and the small scale vertical magnetic flux tubes. The authors analyzed the relation between SST data and HMI LOS magnetogram data, but found only 3.3% clear relation between them as shown in Figure 1 of the response to the reviewers comment. The present referee is positive for the publication of the basic results on the discovery of many swirls in the chromosphere and photosphere and their intimate relation. However, the relation between swirls and vertical magnetic flux tubes are remained to be examined in detail in future. Hence I would like to recommend the authors to publish all results including the relation to HMI LOS magnetogram data.

Reviewer #3 (Remarks to the Author):

While the revised manuscript addresses some of my concerns in the previous report, major concerns still remain. Therefore, unfortunately, I cannot recommend publication in Nature Communications.

In particular, the study does not adequately address the poor correlation of the swirls detected at various solar atmospheric layers. The discussion of the energy shortage in the detected waves is contradictory by first claiming significant energy and then indicating a required energy for mid-chromospheric heating that is 3-10 times larger than the estimated wave energy flux.

The paper does not provide adequate detail on the numerical method and parameters used in the present study, which does not allow assessing the numerical adequacy of the model.

Response to Reviewers' Comments

Reviewer #1 (Remarks to the Author):

This is a revised version of a previously submitted manuscript. The original manuscript contained strong claims about propagating Alfvén pulses found to carry enough energy to sustain the high temperatures of quiet regions in the entire upper atmosphere. These claims were not supported by the presented evidence.

The authors have now lessened their claims and present a fair manuscript where the finding and properties of photospheric and chromospheric short-lived swirls and their statistical relationship are presented.

An estimate about the energy potentially to be carried by these events is presented. The magnitude is high enough for these events to be considered important in the energy transport across the solar atmosphere.

The authors employ state-of-the-art instrumentation with data of the highest available quality. A multi wavelength multi instrument analysis is performed. Also, advanced numerical computations were carried out trying to mimic the generation and propagation of the disturbances thought to be responsible for the observed swirl propagation.

I find these results are certainly of interest to the solar community and perhaps also to readers from other fields. The claims are now in accordance with the level of evidence being presented. For these reasons, I recommend the manuscript for acceptance in Nature Communications.

Response 1: We appreciate the reviewer thinking highly of our work and recommending the acceptance of the article in Nature Communications.

Reviewer #2 (Remarks to the Author):

The authors responded various comments and questions by the referee seriously to some extent, which improved the paper very much. Only one concern is the relation between the swirls and the small scale vertical magnetic flux tubes. The authors analyzed the relation between SST data and HMI LOS magnetogram data, but found only 3.3% clear relation between them as shown in Figure 1 of the response to the reviewers comment. The present referee is positive for the publication of the basic results on the discovery of many swirls in the chromosphere and photosphere and their intimate relation. However, the relation between swirls and vertical magnetic flux tubes are remained to be examined in detail in future. Hence I would like to recommend the authors to publish all results including the relation to HMI LOS magnetogram data.

Response 2: We thank the reviewer for the recommendation of the publication of our work. We have now included the analysis of the relation between SST photospheric swirls and HMI LOS magnetic field data in the Supplementary Material (see the section in purple titled “Possible Relationship with Small-scale Magnetic Flux Tubes” and Fig. S7).

Reviewer #3 (Remarks to the Author):

While the revised manuscript addresses some of my concerns in the previous report, major concerns still remain. Therefore, unfortunately, I cannot recommend publication in Nature Communications.

In particular, the study does not adequately address the poor correlation of the swirls detected at various solar atmospheric layers.

Response 3.1: We highly appreciate this comment which has led us to study further into the correlation between photospheric and chromospheric swirls. We would like to make it clear that:

1) The peak correlation index (CI) is indeed low, but as expected. As we stated in the section *Estimation of Correlation Indices and Overlaps* of the **Methods**, the CI was used to find the corresponding time lag where the photospheric and chromospheric swirl detection results match each other the most. However, the CI of two given layers cannot give us information of how many swirls are overlapped between different layers. For example, a CI of 0.01 could either mean only 1% of the swirls in one layer are overlapped with those in the other layer, or all swirls are overlapped but each swirl has only 1% points overlapped, in extreme cases. We would also note that, the peak CI is the average correlation between all photospheric frames and chromospheric frames with a certain time lag. Considering the magnetic field and plasma inhomogeneity in the solar atmosphere, different Alfvén pulses even detected in the same frame are not expected to travel upward at the same speeds, meaning different time lags between the photosphere and chromosphere for different swirls detected at the same time, let alone different swirls detected in different frames at different time instances.

2) Although even a peak overlap of ~41% means already a high number (suggesting a population of 0.66×10^5 expected in the photosphere) of SOT photospheric swirls that have been found to correspond to chromospheric swirls, it is still the result for just a single time lag between photospheric and chromospheric observations. Again, the inhomogeneity of the magnetic field and plasma properties in the solar atmosphere needs to be kept in mind. We have performed further analysis, to check, for example, how many photospheric swirls could be found to be overlapped with chromospheric swirls within a time lag range of 100 to 160 s. Fig. S4(b) in the revised Supplementary material shows the histogram of the percentage of such swirls in each SOT photospheric frame. It is shown that: 1) about 81% of the detected SOT photospheric swirls have correspondences in the chromosphere within a time lag range of 100 s to 160 s; and 2) about 94% of the photospheric frames have more than half swirls overlapping with chromospheric swirls found within a time lag range of 100 s to 160 s. Similarly (Supplementary Fig. S5(d)), about 86% of the detected SST Fe I photospheric swirls have correspondences in the Ca II 8542 Å line core chromospheric observations within a time lag range of 100 s to 160 s; and 2) about 99% of the SST photospheric frames have more than half swirls overlapping with chromospheric swirls found within a time lag range of 100 s to 160 s.

We have now added the above discussions into the article (see line 80 - 89 in the main article).

The discussion of the energy shortage in the detected waves is contradictory by first claiming significant energy and then indicating a required energy for mid-chromospheric heating that is 3-10 times larger than the estimated wave energy flux.

Response 3.2: Despite all the effects we have listed in page 4 of the Supplementary Material, which show that the estimated average energy flux carried by these Alfvén pulses should be the lower limit, we agree with the referee that we need more evidence to show their exact contribution to the global heating of the upper chromosphere. We have now further weakened the statement in the abstract and conclusions as “*The estimation of the energy flux carried by these Alfvén pulses (Supplementary text) suggests that these abundant events are potentially able to carry a considerable amount of non-thermal energy (1.9 to 7.7 kW m⁻²) into the upper chromosphere. Estimates suggest that this energy flux is enough to balance the local radiative (and other) energy losses (of the order of 0.1 kW m⁻²) in quiet regions. The exact contribution of these Alfvén pulses to the global upper chromospheric heating needs to be further studied by applying ASDA to observational data with higher spatial and temporal resolutions.*” Please find the changes as texts in purple.

The paper does not provide adequate detail on the numerical method and parameters used in the present study, which does not allow assessing the numerical adequacy of the model.

Response 3.3: Thank you for pointing out that more details would be needed on the numerical method and parameters used. We have now added the requested details, including assumptions and parameters used in the simulation, into the **Methods** (see texts in purple) and the Supplementary Material (see the new section titled *Governing Equations of the Numerical Simulation and the Construction of the Single Magnetic Flux Tube*). In **Methods**, besides the newly added information on the spatial resolution of the simulations, we have also added the following to justify its use:

“SAC separates background and perturbation variables in order to accurately resolve perturbations in a stratified atmosphere. This approach is suitable when the total variations of the background plasma are small compared to the perturbation value, for example, waves propagating along a flux tube. SAC has been tested and used to study numerically such propagations previously (e.g., Fedun et al. 2011)”

Under the section of *Governing Equations of the Numerical Simulation and the Construction of the Single Magnetic Flux Tube* in the Supplementary Material, we have now added the governing equations of the simulations, together with the applied assumptions and constants used. We have also added references where interested readers could find more information on the implemented numerical diffusion and resistivity terms. We have now also addressed in details the analytical model used to construct the open magnetic flux tube. Besides the governing equations, the characteristic parameters used including e.g. the footpoint magnetic field strength (800 G), the radial scaling factor (39.38 km), and the chromospheric scale height (0.45 Mm) are also provided. We have further given the reference for the approach used to model the pressure balance for readers to access.

Reviewers' comments:

Reviewer #2 (Remarks to the Author):

I quickly check the revised version, especially for the part I recommended to revise, and found that the authors revised and added the negative results honestly on the relation between swirls and vertical magnetic flux tubes because of lack of spatial resolution of HMI in supplementary material.

Then the reader can check the reliability of the paper on the basis of these results.

Hence I now recommend publication of this paper in the present form.

Reviewer #3 (Remarks to the Author):

The authors have addressed some of my comments of the previous report. However, a couple of serious concerns remains.

The authors claim that the Alfvén pulses propagate upwards through the solar atmosphere carry $1.9-7.7 \text{ kW/m}^2$ energy flux and now compare it to the average loss of quiet chromospheric regions of 0.1 kW/m^2 . In the present version paper and in previous version the authors have calculated the average energy flux carried by these waves is $0.033-0.131 \text{ kW/m}^2$ (line 84 of the supplement), and the average flux is more appropriate value to compare to the average quiet chromospheric losses. I find the present description misleading in that one has to look carefully in the supplement in order to identify the relevant energy flux that may balance chromospheric losses. Also, it is still not evident

in the present study whether enough wave energy flux is actually dissipated to provide the required heating.

The authors use the WKB approximation in order to estimate the wave energy flux (equation S3). The WKB approximation neglects reflection and dissipation in the calculation of the energy flux. Thus, the energy flux could be in fact much lower, if one accounts for the above effects. This should be clearly noted and discussed in the supplement.

In summary, given the estimated marginal energy flux provided by the Alfvén pulses for chromospheric heating, and still uncertain effects of dissipation and reflection in the present study, I am not convinced that the claims made in the paper are more than speculative, and of interest to the general scientific community. Therefore, I still cannot recommend publication in Nature Communications.

Independent reviewer's remarks to the author:

I would be happy if the authors can explicitly state the limitations raised by Reviewer 3 in their manuscript. In particular,

Regarding the comment:

- I find the present description misleading in that one has to look carefully in the supplement in order to identify the relevant energy flux that may balance chromospheric losses. Also, it is still not evident in the present study whether enough wave energy flux is actually dissipated to provide the required heating.

The authors should shortly discuss this in the main text.

Regarding the comment:

The authors use the WKB approximation in order to estimate the wave energy flux (equation S3). The WKB approximation neglects reflection and dissipation in the calculation of the energy flux.

Thus, the energy flux could be in fact much lower, if one accounts for the above effects. This should be clearly noted and discussed in the supplement.

The authors should explicitly mention this limitation in the supplement.

Response to Reviewers' Comments

Comment 1:

From Reviewer 3: The authors claim that the Alfvén pulses propagate upwards through the solar atmosphere carry $1.9\text{--}7.7\text{ kW/m}^2$ energy flux and now compare it to the average loss of quiet chromospheric regions of 0.1 kW/m^2 . In the present version paper and in previous version the authors have calculated the average energy flux carried by these waves is $0.033\text{--}0.131\text{ kW/m}^2$ (line 84 of the supplement), and the average flux is more appropriate value to compare to the average quiet chromospheric losses. I find the present description misleading in that one has to look carefully in the supplement in order to identify the relevant energy flux that may balance chromospheric losses. Also, it is still not evident in the present study whether enough wave energy flux is actually dissipated to provide the required heating.

From the Independent Reviewer on “I find the present description misleading in that one has to look carefully in the supplement in order to identify the relevant energy flux that may balance chromospheric losses. Also, it is still not evident in the present study whether enough wave energy flux is actually dissipated to provide the required heating.” The authors should shortly discuss this in the main text.

Response 1: Concerning “I find the present description misleading in that one has to look carefully in the supplement in order to identify the relevant energy flux that may balance chromospheric losses.” Besides the descriptions we have already had in the main text on the estimation of the energy flux carried by a single Alfvén pulse (Line 114 to 117), we have now added the following (text in purple in Line 117) into the main text: “The average energy flux contribution by these Alfvén pulses is estimated to be 33 to 131 W m^{-2} (see Supplementary text).”

Concerning “Also, it is still not evident in the present study whether enough wave energy flux is actually dissipated to provide the required heating.” We are sorry that, Reviewer 3 had not raised this concern in his/her previous reports at all. This concern was raised by Reviewer 1, and we guess Reviewer 3 may not remember the questions being raised by another reviewer, and may have missed the corresponding responses we provided. Our response to this particular concern, quoted from our previous letter, was as follows:

We agree that the dissipation mechanism is indeed a very important part in solving the heating problem. However, we would like to emphasise that it is also a completely different topic from what we are currently aiming to address in the paper. As far as we are aware of, the damping mechanism of MHD waves has been majorly underpinned by energy cascade from large to small scales. There are few different theories trying to explain the dissipation mechanism of MHD waves in the solar atmosphere: 1) mode coupling (e.g., Pascoe et al. 2013); and 2) instability or turbulence (e.g., Parker 1991; Oughton et al. 2001; Cranmer & van Ballegoijen 2005; Verdini et al. 2010, just to name a few from the vast literature on this topic). However:

- 1) There is still an intense debate about the applicability of these different heating mechanisms to dissipate the transported energy effectively.*
- 2) Most papers on the heating mechanisms have been focused on theories or numerical simulations. Until now, no solid direct and decisive observations have been provided to the community of which mechanism is ubiquitous and is playing the most important role in dissipating wave energy in the solar atmosphere.*
- 3) The size of swirls we have found in our paper is already at the limit of current resolution, and it is impossible to directly observe the cascade to even smaller scales with the current facilities.*

Based on the above facts and indeed that our supporting simulation was not designed for tackling the dissipation problem, we would not wish to make any conjecture or speculations on the dissipation mechanism of the observed Alfvén pulses. We do not know the exact heating mechanism yet, any conjecture without solid support from direct observations may mislead the community and affect the scientific reliability of our paper.

We would like to stress that, the purpose of this paper, is to provide direct and solid evidence of the presence of Alfvén pulses, which could carry a considerable amount of energy and have potential abilities to heat the quiet upper chromosphere, propagating from the photosphere to the chromosphere. We, here, did not intend to solve the other problem of how the waves/pulses would dissipate their energy in the solar atmosphere, just as many other observations (to name a few, Wedemeyer et al. Nature 2012; Martínez-Sykora et al. Science 2017) did not tackle this problem either. Dissipation is a related but completely different problem.

Having taken the above into account, in Lines 120-122 of our manuscript we had already stated that “Dissipation of these Alfvén pulses has not been accounted for in the simulation, given the limited ability to resolve small-scale instabilities and turbulence. This important topic should be further studied in the future to illustrate how the energy carried by these pulses would be deposited to heat the local upper chromosphere.”

Comment 2:

From Reviewer 2: The authors use the WKB approximation in order to estimate the wave energy flux (equation S3). The WKB approximation neglects reflection and dissipation in the calculation of the energy flux. Thus, the energy flux could be in fact much lower, if one accounts for the above effects. This should be clearly noted and discussed in the supplement.

From the Independent Reviewer on the above: The authors should explicitly mention this limitation in the supplement.

Response 2: We would like to point out that the WKB approximation itself does not neglect the reflection or dissipation. The WKB approximation deals with the propagation of MHD waves under the short-wavelength assumption (e.g., Mattaeus et al. 1994). However, the Alfvén energy flux calculated in the supplementary material was the energy flux at a single surface at the bottom of the upper chromosphere - no propagation was involved. Even though the classical WKB theory relies on the small-amplitude approximation, it can be extended to large-amplitude waves at least for the Alfvén mode (e.g., Barnes and Hollweg 1974, Barnes 1992). In this case, the WKB approximation is irrelevant.

However, equation S3 is indeed based on the small-amplitude assumption, so we have now revised Line 73 to 74 in the Supplementary Text as: “Under the small-amplitude assumption, the energy flux carried into the upper chromosphere by a single Alfvén pulse could be expressed as”

Concerning that the energy flux could be affected by the possible reflection and dissipation, we have now added the following to Lines 105-113 in the Supplementary Text: We shall also note that, possible reflection and dissipation may also affect the local energy budget. However, we do not see any evidence of reflection or dissipation of these Alfvén pulses in the data. This may be an interesting future direction to investigate, but is beyond the scope of the current study. Moreover, the estimation of the energy flux of the observed Alfvén pulses is based on either empirical/theoretical (mass density and Alfvén speed) or observational (the average rotating speed of swirls) results at the bottom of the upper chromosphere, thus possible reflection and dissipation during their propagation from the photosphere to the bottom of the upper chromosphere are irrelevant in the above energy flux estimation.

REVIEWERS' COMMENTS:

Reviewer #3 (Remarks to the Author):

The authors have addressed most of my comments and concerns, and I am now happy recommending publication after the following minor corrections/edits:

1. In line with the authors responses and statements on lines 117-122 they should mention this limitation of their study in the abstract after the current last statement: "It should be noted that, whether this wave energy flux is actually dissipated in the chromosphere and leads to heating that balances the losses is still an open question."
2. In the Supplement line 73: "Under the small-amplitude assumption" => "Under the small-amplitude, short wavelength assumption"

Response to Reviewers' Comments

Comment 1:

In line with the authors responses and statements on lines 117-122 they should mention this limitation of their study in the abstract after the current last statement: "It should be noted that, whether this wave energy flux is actually dissipated in the chromosphere and leads to heating that balances the losses is still an open question."

Response 1: We have now added the following to the end of the abstract "Whether/how this energy flux is dissipated and leads to the heating is still an open question."

Comment 2:

In the Supplement line 73: "Under the small-amplitude assumption" => "Under the small-amplitude, short wavelength assumption"

Response 2: Now revised.